# A new global dataset of mountain glacier centerline and length
**Dahong Zhang [1,2], Gang Zhou [1,2], Wen Li [1,2], Shiqiang Zhang [1,2], Xiaojun Yao [3], Shimei Wei [3]**
[1] College of Urban and Environmental Science, Northwest University, Xi'an 710127, PR China
[2] Shaanxi Key Laboratory of Earth Surface System and Environmental Carrying Capacity, Northwest University,
Xi'an 710127, PR China
[3] College of Geography and Environment Sciences, Northwest Normal University, Lanzhou 730070, PR China
*Correspondence*: Shiqiang Zhang (zhangsq@lzb.ac.cn)
**Abstract.** Length is one of the key determinants of glacier geometry and is an important parameter
of glacier inventory and modeling; glacier centerlines are crucial inputs for many glaciological
applications. In this study, the centerlines and maximum lengths of global glaciers were extracted
using an automatic extraction algorithm based on the latest global glacier inventory data, digital
elevation data (DEM), and European allocation theory. The glacier polygons were reconstructed
according to the geometric principle and an automatic checking algorithm for the global glacier
outlines was designed to filter erroneous or unsupported glacier outlines. The DEMs of global
glacier-covered regions were compiled using available DEMs. An updated automatic extraction tool
was designed independently, and a parameterization scheme with empirical thresholds was applied
for data production. The accuracy of the dataset was evaluated using random assessment with visible
interpretation and comparative analysis with another dataset. The 10,764 erroneous glacier polygons,
7,174 ice caps, and 419 nominal glaciers from the Randolph Glacier Inventory (RGI) version 6.0
were identified and excluded, accounting for 8.25% of the total. In total, 198,137 glacier centerlines
were generated, accounting for 99.74% of the total input glaciers and 91.52% of the RGI v6.0. The
accuracy of glacier centerlines was 89.68%. The comparison between the dataset and previous
datasets suggested that the majority of glacier centerlines were slightly longer than those in RGI
v6.0. The extraction method of this study has a strong ability to obtain the maximum length of
glaciers, meaning that the maximum lengths of some glaciers were likely underestimated in the past.
The dataset constructed includes 14 sub-datasets, such as the global glacier centerline dataset, global
glacier maximum length dataset, and global glacier DEM dataset, all of which can be found at link:
https://doi.org/10.11922/sciencedb.01643 (Zhang and Zhang, 2022).
## 1 Introduction
Mountain glaciers which are distinct from the Greenland and Antarctic ice sheets, are also shrinking
rapidly (Hugonnet et al., 2021). They are altering regional hydrology (Pritchard, 2019), raising
global sea levels (Cazenave, 2018), and elevating natural hazards (Shukla and Sen, 2021; Zheng et
al., 2021). These glaciers are among the most climate-sensitive constituents of the world's natural
water towers (Immerzeel et al., 2019). Under the influence of global climate change, studies on
glacier area changes(Sommer et al., 2020; Li et al., 2021), ice thickness (Farinotti et al., 2019), mass
balance (Zemp et al., 2019; Vargo et al., 2020; Wu et al., 2021), ice velocity field (Thogersen et al.,
2019), the impact of debris-cover (Scherler et al., 2018; Shukla et al., 2020; Herreid and Pellicciotti,
2020), glacier meltwater (Noel et al., 2020), sediment release (Aciego et al., 2015; Li et al., 2019),
and related hazards (Zhou et al., 2021b; Stuart-Smith et al., 2021; Kääb et al., 2021) in glacier-
covered regions are essential for global water resources supply and disaster prevention and reduction.



The most obvious distinction between glaciers and other natural ice bodies is their property to move
towards lower altitudes under the influence of gravity. Glacier flow lines are the motion trajectories
of a glacier and the main flow line is the key trajectory. The main flow line cannot be obtained on a
large scale owing to the lack of glacier velocity field data. The glacier centerline, generated via the
axis line method (Le Bris and Paul, 2013; Machguth and Huss, 2014; Kienholz et al., 2014; Zhang
et al., 2021), is typically used to represent the main flow line. The glacier centerline is a critical
parameter for analyzing the ice velocity field (Heid and Kääb, 2012; Melkonian et al., 2017),
estimating the glacier volume (Li et al., 2012; Gao et al., 2018), and developing glacier models
(Oerlemans, 1997; Sugiyama et al., 2007; Maussion et al., 2019).
Glacier length, usually referring to the maximum length of a glacier centerline (main flow line),
represents the longest motion trajectory of a glacier, which is one of the key determinants of glacier
geometry and a basic parameter of glacier inventories (RGI Consortium, 2017) and modeling
(Maussion et al., 2019). Glacier length fluctuations can be used to quantify glacier changes (Zhou
et al., 2021a), such as by identifying glacier advancement, surge, or retreat. Glacier length
fluctuations (e.g., Leclercq et al., 2014) have also been used to study the relationships with changes
in glacier area (Winsvold et al., 2014) and the geometric structure of a glacier (Herla et al., 2017),
estimate glacier volume in combination with the glacier area (Lüthi et al., 2010), and reconstruct
annual averaged surface temperatures over the past 400 years on hemispherical and global scales
(Leclercq and Oerlemans, 2011).
The global complete inventory (RGI Consortium, 2017) of glacier outlines was created following
the Fifth Assessment Report of the Intergovernmental Panel on Climate Change (IPCC AR5). To
meet the demand for large-scale acquisition of glacier length, automatic and semi-automatic
methods have been proposed. There are three types of methods: first, the typical hydrological
analysis method (Schiefer et al., 2008), but the lengths are longer than equivalent maximum
distances taken along typical longitudinal centerline profiles; second, a simplified algorithm based
on skeleton theory (Le Moine and Gsell, 2015), but this method has not been widely used; third,
centerline method based on the axis concept, proposed by Le Bris and Paul (2013), and applied to
the calculation of global glacier length for the first time by Machguth and Huss (2014). However, it
is difficult to extract complex glaciers using the method of (Le Bris and Paul, 2013). The cost grid-
least-cost route approach of Kienholz et al. (2014) based on the axis concept has higher accuracy,
but it requires more labor and time, which limits its application to global glaciers. The trade-off
function approach of Machguth and Huss (2014) was based on the axis concept; the results cover
almost all mountain glaciers in the world but exclude the centerlines of branches of glaciers.
Therefore, researchers have been trying to overcome these difficulties in recent years (Yao et al.,
2015; Yang et al., 2016; Ji et al., 2017; Hansen et al., 2020; Xia, 2020; Zhang et al., 2021). To date,
global datasets of the centerline and length of mountain glaciers are rare, including that of glacier
branches. Based on our recent study (Zhang et al., 2021) on successfully extracting the glacier
centerline using the Euclidean allocation, in this study, we aim to combine free, available digital
elevation data into one global digital elevation model (DEM) with 30 m resolution from 90°N to
90°S, check and correct the global glacier outlines, and obtain a new graphic dataset of the centerline
and length for global mountain glaciers based on the updated DEM and outlines.

## 2 Study region and data

The glacier dataset used in this study was the RGI version 6.0 (http://www.glims.org/RGI/randolph.html, last accessed: 15 November 2021) released via the Global Land Ice Measurements from Space initiative (GLIMS), which is a globally complete collection of digital outlines of glaciers, excluding ice sheets (Pfeffer et al., 2014). RGI v6.0 includes 216,502 glaciers (215,547 glaciers described in the product handbook) worldwide, with a total area of 705,738.793 km$^2$ (RGI Consortium, 2017). All glaciers can be divided into 19 first-order glacier regions (Radić and Hock, 2010), and these regions were used in our study (Fig. 1).

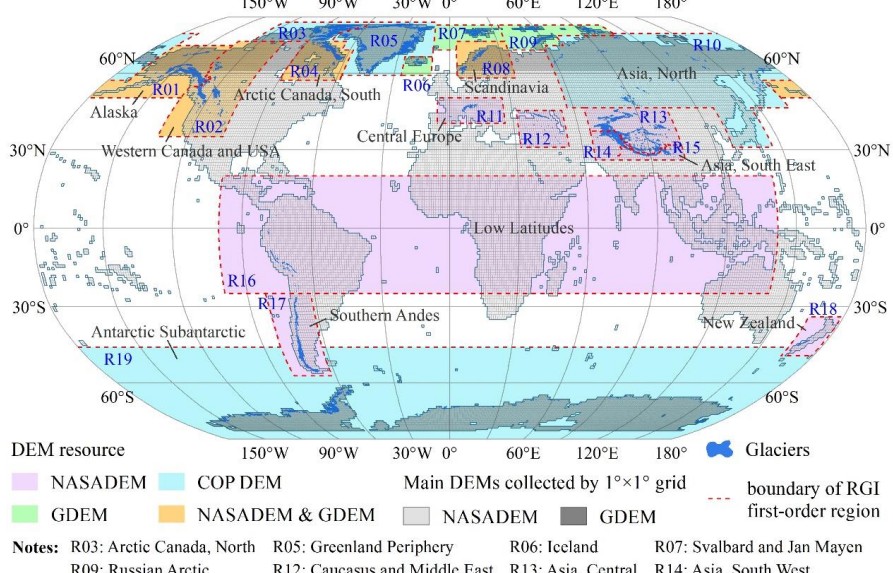

**Figure 1.** Distribution of global glaciers, first-order glacier regions, and DEMs. The background is the global DEM grid (1°×1°) covered by NASADEM and GDEM. GDEM and COP DEM represent the ASTER GDEM v3 and the Copernicus DEM, respectively.

Five DEM products (Table 1) were collected in preliminary studies. The National Aeronautics and Space Administration (NASA) DEM (NASADEM) (https://lpdaac.usgs.gov/news/release-nasadem-data-products/, last accessed: November 17, 2021) was released by the Land Processes Distributed Active Archive Center (LP DAAC) in January 2020. As a modernization of the DEM and associated products generated from the Shuttle Radar Topography Mission (SRTM) data (Farr et al., 2007), the NASADEM, with a low mean absolute error (MAE) (Carrera-Hernández, 2021), is the successor of the NASA SRTM V3. The root mean square error (RMSE) of NASADEM is better than that of SRTM (Uuemaa et al., 2020). Serving the zonal extent of (56°S, 61°N), NASADEM was used as the preferred DEM in this study because of its superior performance. The Advanced Spaceborne Thermal Emission and Reflection Radiometer (ASTER) is a 14-channel imaging instrument operating on the Terra satellite of NASA since 1999. ASTER Global Digital Elevation Model (GDEM) version 3 (https://lpdaac.usgs.gov/news/nasa-and-meti-release-aster-global-dem-version-3/, last accessed: November 17, 2021) (Abrams et al., 2020) was released by Japan's Ministry of Economy, Trade, and Industry (METI) and NASA in July 2019. Using ICESat



data, Carabajal and Boy (2016) found that ASTER GDEM v3 displayed smaller means, similar
medians, and less scatter than ASTER GDEM v2 in Greenland and Antarctica. To determine the
zonal extent of (56°S, 83°S) and (61°N, 83°N), ASTER GDEM v3 was used as the second priority
DEM in this study.
NASADEM and ASTERGDEM v3 do not cover all glacierized regions, missing parts of the polar
region and the Kamchatka Peninsula. Because of their high temporal and spatial resolution at high
latitudes, the reference elevation model of Antarctica (REMA) (Howat et al., 2019) and ArcticDEM
(https://www.pgc.umn.edu/data/arcticdem/, last accessed: November 17, 2021) were preferred as
the supplementary data of our preliminary studies in these glacier regions. However, ArcticDEM
and REMA are not suitable because of insufficient coverage and sporadic data. Therefore,
Copernicus DEM (https://spacedata.copernicus.eu/web/cscda/cop-dem-faq, last accessed:
November 17, 2021) with a wide coverage was finally determined as the supplementary data for
glacier regions not covered via the NASADEM and ASTER GDEM v3 completely. The Copernicus
DEM was recently released (November 2020) and the accuracy assessment undertaken by its
development team (the product handbook) using TanDEM-X DEM/World DEM ICESat GLAS
reference points found an absolute vertical accuracy of approximately 10 m at the periphery of
Antarctica and Greenland. Finally, NASADEM, ASTER GDEM v3, and Copernicus DEM were
compiled to create a 30 m DEM of the completely covered study area.

**Table 1.** All DEMs collected in this study

| DEM | Extent | Resolution | Access |
|---|---|---|---|
| NASADEM | [56°S, 61°N] | 30 m | https://search.earthdata.nasa.gov/search |
| ASTER GDEM v3 | [83°S, 83°N] | 30 m | https://gdemdl.aster.jspacesystems.or.jp/ |
| ArcticDEM | [55°N, 90°N] | 2 m | https://earthengine.google.com/ |
| REMA | [60°S, 88°S] | 2m / 8m | https://earthengine.google.com/ |
| Copernicus DEM | Global | 30 m | https://panda.copernicus.eu/web/cds-catalogue/panda |

**Note:** The interval in the 'Extent' column represents all landmasses within a zonal range, but coverage may not exist
for all areas.
In addition, graphical data (Machguth and Huss, 2014) of glacier length in *. xy format with an
unknown projection coordinate system in High Asia were collected, which correspond to the
attribute of the glacier maximum length ($L_{max}$) in RGI v6.0. Because the data was obtained from an
unofficial source, we could not access the data description documents and only recovered the
coordinate matching between these points and some glaciers in RGI v6.0. Registration of the *. xy
file depends on the match between its filename and the feature identity document (FID) of the glacier
polygon of RGI v6.0 in the same glacier area. The glacier lengths ($MHMLDS$) of successful
registration were used as graphical validation data for this study.
**3 Methods**
**3.1 Outline of workflow**
This study relied on two key input datasets: global glacier inventory and compiled global glacier
elevation. The goal of this study was to establish a new dataset of global graphic glacier centerlines
and lengths. An outline of the workflow is shown in Figure 2. The process was divided into six parts:
(1) design an algorithm to check all glacier outlines, marks, and exclude defective glacier polygons;
(2) buffer glaciers to produce a mask containing global glaciers and their buffers; (3) mosaic





compiled global DEMs according to the masks in step 2 of different glacier regions to prepare the
global glacier elevation data; (4) determine the automatic extraction parameters of glacier
centerlines around the world by repeated testing in each region; (5) input the global DEM and glacier
outline dataset and all parameters into the designed automatic extraction software (Zhang et al.,
2021) to generate the centerlines and length in the global glacier; and (6) verify and compare with
other centerline results obtained via different methods to evaluate the accuracy of the new datasets.

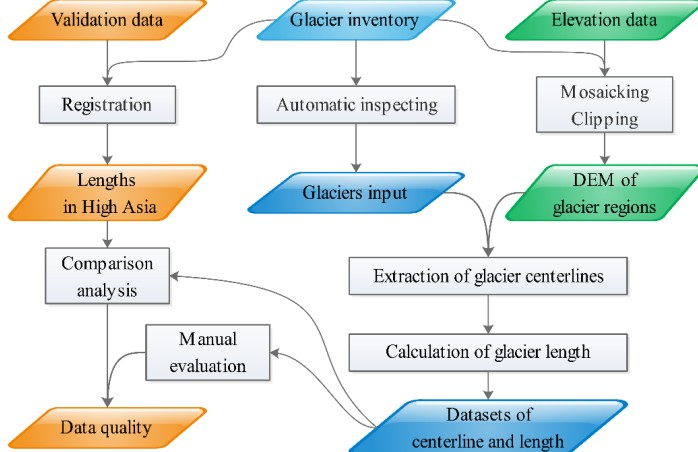


**Figure 2.** Workflow of the centerline and length dataset production.


**3.2 Illustration of key methods**
**3.2.1 Pre-process of glacier outlines**
This study had strict requirements for glacier outlines, and all glacier complexes should be divided
into individual glaciers before centerline extraction. However, because of the limited semi-
automatic glacier segmentation approach (Kienholz et al., 2013) and the high-priority strategy of
completeness of coverage adopted by RGI v6.0 (RGI Consortium, 2017), some glaciers were not
supported by our algorithm. These glaciers included three categories: glacier complexes
with/without inaccurate segmentation (Fig. 3a-b), incorrect glacier outlines (Fig. 3c), and flawed
glaciers (Fig. 3d-f) generated by the automatic extraction algorithm. For the third category, we
designed an identification algorithm (see paragraph 3) to mark and screen them. The flaws in these
glacier outlines were mainly caused by topology errors of polylines/polygons, such as unclosed,
sawtooth, and overlap. The first two categories do not affect the program's normal operation;
however, the accuracy of the extraction results is difficult to guarantee. We cannot identify them at
present and a solution is needed to improve the quality of the global glacier inventory.

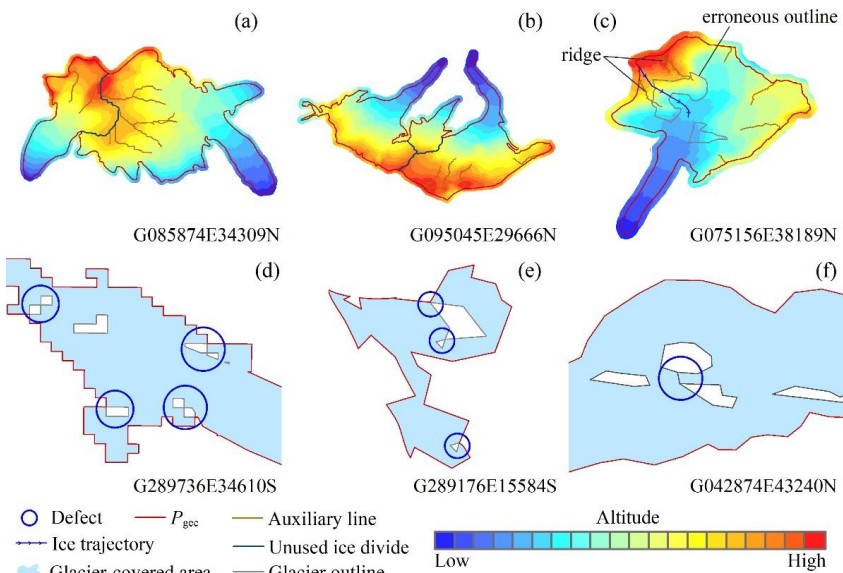

**Figure 3.** Schematic of three types of flawed glacier outlines. **(a-b)** Glacier complexes with/without inaccurate segmentation. **(c)** Incorrect glacier outline. **(d-f)** Panels represent three common problems in flawed glaciers: defects in automatic algorithm, defects in post-processing, and artificial errors. Auxiliary line represents lower-grade ice divide in the individual glacier, which is part of the ridge lines.

In this study, we defined the external contour of a glacier ($P_{gec}$), namely, the polygon corresponding to the longest closed polyline of the glacier, to reduce the storage of DEMs and improve the efficiency of batch processing. The buffer masks of all glaciers (buffer distance: approximately 100 m) were generated by their $P_{gec}$ to meet the requirement for the extent of input DEMs to be slightly larger than the $P_{gec}$. The buffer masks generated initially were relatively broken because there are overlaps or gaps between the adjacent polygons of the buffer zone. We merged small spots to remove polygons with a perimeter less than 12 times the buffer distance on the glacier buffer masks of each region.

The third category of glaciers (Fig. 3d-f) with flaws was identified by obtaining $P_{gec}$. In the third category, the most common type is two or more closed polylines with the same endpoint in a glacier. There were also a few glacier polygons with false closed polylines, which are the head and tail endpoints of the polylines that do not coincide, but the distance is less than the tolerance. The solution was as follows: flawed glacier outlines were identified by judging whether there were multiple polylines sharing endpoints after converting the glacier from a polygon to polylines, but these outlines do not include the false closed type.

### 3.2.2 Preparation of input datasets

All data were processed in units of first-order glacier regions. The input glacier outlines excluded all the defective glacier outlines. Similarly, the nominal glaciers (represented by an ellipse) and ice





caps remarked in RGI v6.0 were also treated, which were distinguished by two attributes: status
(nominal glacier) and form (ice cap). The inspection results (Table 2) of glacier outlines show that
there are 10,764 defective glacier outlines (*FGODS*) in RGI v6.0, accounting for approximately
4.97% of the total (216,502). After excluding nominal glaciers (461) and ice caps (7,174), 198,646
glaciers remained as input glacier outlines (*IGODS*), accounting for 91.75% of the total global
mountain glaciers.
**Table 2.** Preprocessing results of different glacier regions and information of input datasets.

| Region | Region Name | Total | Ice Cap | Nominal glacier | Flawed glacier | Glacier input | DEM input |
|---|---|---|---|---|---|---|---|
| R01 | Alaska | 27108 | 0 | 0 | 704 | 26404 | NASADEM, GDEM |
| R02 | Western Canada and USA | 18855 | 0 | 0 | 1564 | 17291 | NASADEM, GDEM |
| R03 | Arctic Canada, North | 4556 | 650 | 0 | 47 | 3869 | COP DEM |
| R04 | Arctic Canada, South | 7415 | 953 | 0 | 63 | 6409 | NASADEM, GDEM |
| R05 | Greenland Periphery | 20261 | 1658 | 0 | 1547 | 17247 | COP DEM |
| R06 | Iceland | 568 | 133 | 0 | 1 | 435 | GDEM |
| R07 | Svalbard | 1615 | 144 | 0 | 12 | 1460 | GDEM |
| R08 | Scandinavia | 3417 | 0 | 4 | 75 | 3338 | NASADEM, GDEM |
| R09 | Russian Arctic | 1069 | 460 | 0 | 0 | 609 | GDEM |
| R10 | North Asia | 5151 | 5 | 116 | 136 | 4899 | COP DEM |
| R11 | Central Europe | 3927 | 0 | 2 | 76 | 3849 | NASADEM |
| R12 | Caucasus Middle East | 1888 | 0 | 339 | 2 | 1547 | NASADEM |
| R13 | Central Asia | 54429 | 1545 | 0 | 28 | 52858 | NASADEM |
| R14 | South Asia West | 27988 | 295 | 0 | 1946 | 25792 | NASADEM |
| R15 | South Asia East | 13119 | 289 | 0 | 4 | 12826 | NASADEM |
| R16 | Low Latitudes | 2939 | 0 | 0 | 724 | 2215 | NASADEM |
| R17 | Southern Andes | 15908 | 623 | 0 | 3828 | 11734 | NASADEM |
| R18 | New Zealand | 3537 | 0 | 0 | 0 | 3537 | NASADEM |
| R19 | Antarctic Subantarctic | 2752 | 419 | 0 | 7 | 2327 | COP DEM |
| -- | -- | 216502 | 7174 | 461 | 10764 | 198646 | -- |

**Note:** GDEM and COP DEM represent ASTER GDEM v3 and Copernicus DEM, respectively.

$P_{\text{gec}}$ of all glaciers in RGI v6.0 constitute the global glacier external contour dataset (*GGECDS*),
which generated the buffer mask dataset (*GGBMDS*) of global mountain glaciers. The collected
DEMs were extracted using *GGBMDS* and 43,035 DEM tiles were generated. They were then
mosaicked according to different first-order glacier regions to generate a global glacier elevation
dataset (*GGEDS*). The details of the two input datasets are presented in Table 2.

**3.2.3 Generation of centerline and glacier length**
The automatic extraction tool of 'GlacierCenterlines_Py27' (Update to version 5.2.1) was used,
which is based on the axis concept and Euclidean allocation (Zhang et al., 2021). The principle is
briefly explained as follows: the highest and lowest points of the external outline of a glacier as two
endpoints were extracted, cells with the equal shortest distances from the cell to both sides were
identified in a glacier polygon, and the line formed by these cells was regarded as the glacier
centerline. The maximum length of a glacier was calculated using an algorithm similar to the critical
path. The updated contents focused on formulating the parameterization scheme (Appendix A: Table
A1) for extracting global glacier centerlines, as well as repairing some newly discovered bugs, such
as a dead cycle in the process of auxiliary line extraction. All glacier outlines included in the *IGODS*
were divided into ten levels (Table 3) using the proportion of cumulative area after ranking the area
of all input glacier polygons from small to large. User-defined Albers with WGS1984 as the



reference ellipsoid were used as a unified projection coordinate system. The central meridian,
standard parallel 1, standard parallel 2, and origin latitude of the different glacier regions were
determined by their spatial extent. The empirical values of the other parameters were determined in
repeated attempts and their values had a significant correlation with glacier scale. The glacier
centerlines generated were merged according to the glacier regions and the graphics and attribute
information of glacier length were exported as corresponding independent ESRI shapefiles. In
addition, some key associated data were exported, such as the segmentation results of glacier
outlines, the lengths in the accumulation and ablation region of each glacier, the lowest points, the
local highest points ($P_{max}$), the failed glacier outlines dealt, and logs.
**Table 3.** Statistics of global glaciers by different levels.

| Level | Count | Area/km$^2$ | Acc. area/km$^2$ | Percent | Interval/km$^2$ |
|---|---|---|---|---|---|
| L1 | 165593 | 1.00 | 41313.79 | 10% | [0.01, 1.00] |
| L2 | 22833 | 3.57 | 82629.47 | 20% | (1.00, 3.57] |
| L3 | 6906 | 11.39 | 123947.69 | 30% | (3.57, 11.39] |
| L4 | 2149 | 35.51 | 165282.14 | 40% | (11.39, 35.51] |
| L5 | 698 | 103.10 | 206631.32 | 50% | (35.51, 103.10] |
| L6 | 262 | 248.26 | 247917.55 | 60% | (103.10, 248.26] |
| L7 | 113 | 521.40 | 289227.71 | 70% | (248.26, 521.40] |
| L8 | 55 | 1087.47 | 330595.34 | 80% | (521.40, 1087.47] |
| L9 | 27 | 2657.74 | 374312.14 | 90% | (1087.47, 2657.74] |
| L10 | 10 | 6004.85 | 413136.71 | 100% | (2657.74, 6004.85] |
| Total | 198646 | -- | -- | -- | -- |


**3.2.4 Accuracy assessment**
A random assessment was prioritized in this study. We randomly selected 100 glaciers in each
glacier region and obtained 19 samples with a total of 1,900 glacier centerlines. These glacier
centerlines were divided by artificial inspection into three first-level categories (Zhang et al., 2021):
correct (I), inaccurate (II), and incorrect (III). Type II mostly contains glaciers with accurate glacier
maximum lengths but missing, redundant, or unreasonable branches of glacier centerlines. When
calculating verification accuracy, Types I and II were regarded as correct, and only Type III was
considered incorrect. Finally, the glacier proportion of Type III in the sample was counted and the
valuation result ($R$) was calculated using Eq. (1).
$$R = \sum_{i=1}^{19} \frac{S_i \times N_{T_i}}{N_G} \qquad , (1)$$

where $N_G$ is the total quantity of glaciers and $N_{Ti}$ and $S_i$ are the verification accuracy and number of
glaciers in the corresponding glacier region of the $i$ th sample ($i = 1, 2, 3, ..., 18,19$), respectively.
All glacier maximum lengths ($G_{Lmax}$) in this study were compared with the $L_{max}$ (Machguth and
Huss, 2014) in RGI v6.0 using linear correlation and ratio analysis. Here, we took $L_4$ - $L_{10}$ at the
glacier level as the same grade for statistics. The correlations between $G_{Lmax}$ and $L_{max}$ were
established according to different glacier regions and glacier levels and the length ratio, $R_r$ (Eq. 2),
was calculated. In addition, considering the differences between the graphics, we also collected
graph data of glacier length extracted by Machguth and Huss (2014). Considering the limited
availability of data (obtained: R13–R15), we only compared two glacier-covered regions in the
Himalayas: Mount Qomolangma and Kangchenjunga (the world's third-highest mountain) and their
surrounding areas.

Earth System
Science
Data

$R_r = \dfrac{G_{L\max}}{L_{\max}}$     (2)

**4 Results**
**4.1 Centerline and length of glaciers**
Taking the *IGODS*, *GGEDS*, and other model parameters as input data, 198,137 glacier centerlines
were automatically generated using the centerline extraction tool of 'GlacierCenterlines_Py27
v5.2.1', with a success rate of 99.74%. The number and proportion of flawed glacier outlines,
nominal glaciers, ice caps, input glacier outlines, and extraction results for distinct glacier regions
are shown in Fig. 4.

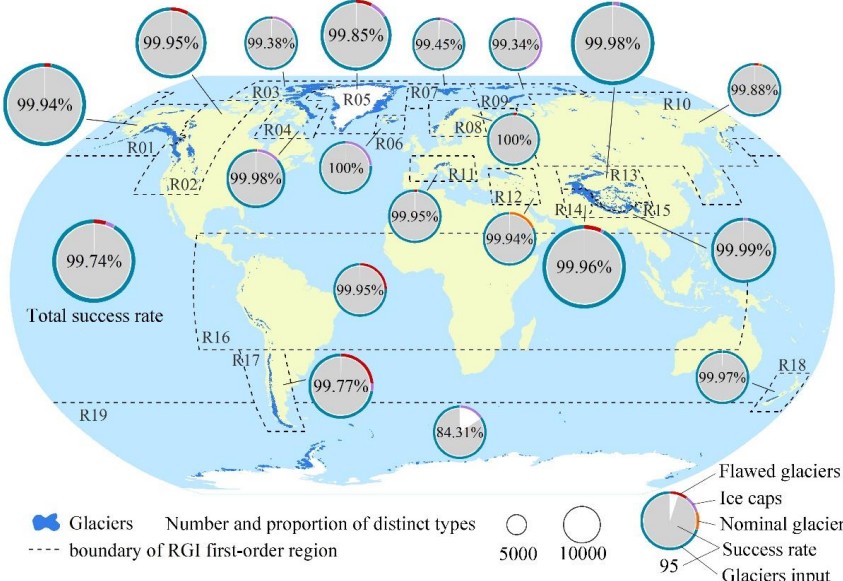


**Figure 4.** Extraction results of glacier centerlines in different glacier regions. The ring in the pie
chart represents the proportion of input glacier number and the number of excluded three glacier
types with total number of glaciers in the region. Pie chart represents the correct rate, which is the
proportion of the extraction result number with input glacier quantity. The size of the pie/ring
represents the grade of the glacier number in the region.

Except for Antarctica and Subantarctica (R19), the success rates of extracting glacier centerlines in
other glacier regions were greater than 99%, which indicates that the automatic extraction algorithm
for glacier centerlines is robust. A small number of glacier outlines with false closed problems and
unidentified ice caps were the main reasons for the failure of automatic extraction of glacier
centerlines; however, it is difficult to establish rules for accurately identifying these glacier polygons.
In total, 510 unsuccessful glacier outlines were identified, of which Antarctic-Subantarctic (R19)
accounted for 71.57%; Southern Andes (R17) and Greenland Periphery (R05) for 5.29% and 5.1%,
respectively; Arctic Canada North (R03) and Alaska (R01) for 4.71% and 2.94%, respectively; and
other glacier regions for less than 2%.






Overall, the global glacier centerline dataset (*GGCLDS*) constructed in this study contained 91.52%
of the total glaciers in RGI v6.0. The lengths of each branch of the glacier centerline were derived
and the longest branch lengths of the glacier centerline were defined as the glacier maximum length
($G_{Lmax}$), which were used to form the global glacier maximum length dataset (*GGMLDS*). The
average centerline length of all branches of a glacier is called the glacier mean length ($G_{Lmean}$). In
addition, the median glacier altitude was regarded as the equilibrium line altitude (ELA) (Machguth
and Huss, 2014), the part with $G_{Lmax}$ higher than ELA was regarded as the length of the glacier
accumulation zone ($G_{Lacc}$), and the part lower than ELA was regarded as the length of the glacier
ablation zone ($G_{Labl}$), which formed the glacier accumulation zonal length dataset (*GACLDS*) and
glacier ablation zone length dataset (*GABLDS*). The key process data corresponding to *GGCLDS*
were also output, to form the glacier outline segmentation results (*GOSRDS*), lowest points
(*GGLPDS*), local highest points (*GLHPDS*), and unsuccessful glacier outlines (*GUGODS*). The
fields involved in all datasets are explained in Table 4.
**Table 4.** Description of the attributes contained in all datasets.

| Name | Data type | Char. length | Description |
|---|---|---|---|
| GLIMS_ID | Char. | 14 | Unique code of a glacier |
| Type | Long int. | 4 | Glacier grade in this study |
| MaxL | Float | 8 | Glacier maximum length (Unit: m) |
| MeanL | Float | 8 | Glacier average length (Unit: m) |
| ELA | Long int. | 4 | Equilibrium line altitude (Unit: m) |
| AccL | Float | 8 | Length in the accumulation region (Unit: m) |
| AblationL | Float | 8 | Length in the ablation region (Unit: m) |
| Id | Long int. | 8 | Data code of the same glacier |
| BS | Long int. | 8 | Tag of the same segment in a glacier |
| RASTERVALU | Long int. | 4 | Altitude of a $P_{max}$ (Unit: m) |

**Note:** Char. and int. represent Character and integer, respectively.

The glacier outlines of RGI v6.0 without centerline results in this study were limited by the quality
of the glacier polygons, which mainly correspond to the flawed glacier outlines *(FGODS)*, and the
identified ice caps in RGI v6.0. Among the *FGODS* (10,764), Southern Andes (R17) had the most,
followed by Southwest Asia (R14); Western Canada and USA (R02) and Greenland Periphery (R05),
with slightly more than 1,500; and Low Latitudes (R16) and Alaska (R01), with slightly more than
700. There were 451 in other glacier regions, including two regions with 0 defective glacier outline,
the Russian Arctic (R09) and New Zealand (R18). Among the ice caps (7174) identified by RGI
v6.0, slightly more than 1,500 were in R05 and Central Asia (R13), between 500 and 1,000 in the
Arctic Canada South (R04), Arctic Canada North (R03), and R17, and less than 500 in other glacier
regions. Nominal glaciers (461) existed in three glacial regions: Caucasus Middle East (R12), North
Asia (R10), and Scandinavia (R08).

**4.2 Data validation**
**4.2.1 Random self-assessment results**
The evaluation results using random samples from the glacier centerline dataset suggested that the
average verification accuracy of the glacier centerline dataset was 89.68%. There were significant
differences in the accuracy of the 19 glacier regions around the world (Fig. 5). Among them, R11,
R15 and R10, R09, and R19 were the highest (98%), second highest (95%), slowest lower (78%),
and lowest (50%), respectively. In terms of types, the average proportions of Types I and II were

Earth System Science Data Discussions Open Access

83.53% and 6.16%, respectively. The proportion of Type I in R07 and R09 was relatively low, at
79% and 73%, respectively, and the lowest in R19 was only 50%. Type II had the highest proportion
in R19 at 16%, followed by R07 (10%). Moreover, Type II accounted for more than 5% in seven
regions: R11, R13, R17, R18, R16, R01, and R06.

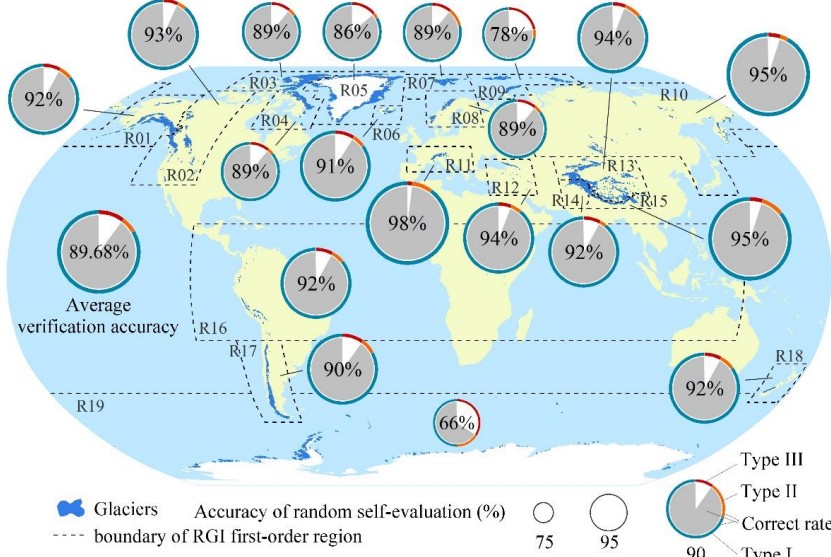


**Figure 5.** Statistical chart of random evaluation results. The ring in the pie chart represents the
proportion of each type with total number of samples in the region. Pie chart represents the correct
rate, which is the proportion of the number of Types I and II with region sample quantity. The size
of the pie/ring represents the grade of the correct rate in the region. Types I, II, and III (See Section
3.2.4) represent the centerline of correct, inaccurate, and incorrect, respectively.


The above results indicate that, in addition to the three glacier regions of R07, R09, and R19, the
random samples of the glacier centerline dataset have excellent performance in terms of accuracy,
particularly in R02, R12, and R14. The unmarked ice cap and local low-quality DEM were the main
reasons for the poor quality of the glacier centerline in R07 and R09, respectively. Owing to glacier
complexes and low altitude differences in low-quality DEMs at the glacier tongue, the quality of
the glacier centerline obtained in R19 was poor. However, from the viewpoint of dataset coverage,
we provided the extraction results of the glacier centerline in R19.

### 4.2.2 Compare with previous results

After applying this algorithm to the global glacier inventory RGI v6.0, we compared the glacier
lengths ($G_{Lmax}$) automatically obtained in this study with those ($L_{max}$) obtained by Machguth and
Huss (2014) (Fig. 6). After eliminating 5408 glaciers with $L_{max}$ value of -9 (no results), the length
values of the other 192728 glaciers in the global glacier length dataset were directly compared. The
$G_{Lmax}$ and $L_{max}$ were generally comparable (Fig. 6a). The glaciers in grades $L_4$–$L_{10}$ showed excellent
fitting degrees, while those of $L_1$–$L_3$ determined the linear correlation coefficient owing to their
large number. The number of glaciers with a length ratio ($R_r$) between $G_{Lmax}$ and $L_{max}$ greater than



1.55 (Fig. 6b) was approximately 35,000, which were excluded from histogram statistics because
there was a high probability that the length of at least one of the two datasets was erroneous. The
peak value of the histogram (Fig. 6b) of $R_r$ is in the interval 1.05–1.15 and $R_r$ in the interval 0.95–
1.25 accounts for 64.55%. The glacier length $G_{Lmax}$ in this study was generally longer than $L_{max}$ and
the average value was approximately 10%, which indicates that glacier centerline lengths were
probably underestimated in previous studies. In addition, the abnormal value of the length ratio of
glacier $L_1$ was the highest and the median value was high (Fig. 6c). The $R_r$ values of glaciers $L_4$–$L_{10}$
fluctuated greatly. The $R_r$ distributions of glaciers $L_2$ and $L_3$ were relatively concentrated. The reason
for this is that the length of glacier $L_1$ was affected by the DEM, while glaciers $L_4$–$L_{10}$ were mainly
disturbed by differences in glacier scale and the accuracy of the auxiliary line.

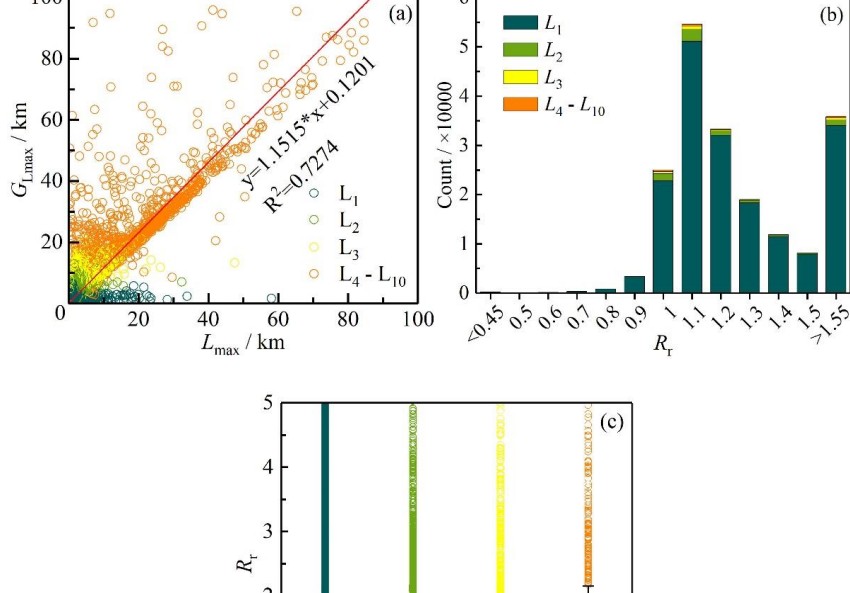



**Figure 6.** Comparison of longest centerlines calculated in this study and by Machguth and Huss
(2014). **(a)** Linear regression of maximum length for all input glaciers (*IGODS*), determined in the
$G_{Lmax}$, calculated in this study and $L_{max}$ obtained in Machguth and Huss (2014). **(b)** Histogram of
length ratio ($R_r$, $G_{Lmax}/L_{max}$) for distinct grades of glaciers. **(c)** Box plots of length ratio ($R_r$) for
different scales of glaciers.

Comparisons between $G_{Lmax}$ and $L_{max}$ for each first-order glacier region and all random samples are
shown in Appendix B. There was a preferable fitting degree between $G_{Lmax}$ and $L_{max}$ in seven glacier
regions including R01, R04, R07, and R12–R15, in which the $R^2$ was larger than 0.95 (Fig. B1).
The $R_r$ in R17 ($R^2 = 0.8174$), R05 ($R^2 = 0.8136$), and R03 ($R^2 = 0.6311$) were poor, whereas that in



R19 ($R^2 = 0.5487$) was the worst. The $R^2$ values of the other eight glacier regions were between 0.85
and 0.95. The histograms (Fig. B2) suggest that $G_{Lmax}$ and $L_{max}$ fitted well in R04, R06, R07, R09,
and R12–R15 because they had recognizable single peak values. The peak values of R03, R05, R17,
and R19 were not prominent and the proportion of glaciers with $R_r > 1.55$ was extremely high,
further increasing the uncertainty in glacier length results in these four regions. R01, R07, R08,
R11–R15, and R18 performed well in the box plot (Fig. B3), whereas the results for R09 were not
good. Moreover, the fitting degree of all random samples was poor (Fig. B1, $R^2 = 0.7547$), the peak
value was more prominent (Fig. B2), and the length ratio distribution of glaciers of different grades
was relatively scattered (Fig. B3). In general, the glacier lengths of R07 and R12–R15 were the
closest, while there were significant differences in R03, R05, R17, and R19.

Furthermore, graphic results collected for the maximum length of glaciers in parts of High Asia
(Machguth and Huss, 2014) were used to compare the results. In two parts of R15, Mount
Qomolangma and its surrounding area (Fig. 7a) and Kangchenjunga and its surrounding area (Fig.
7b), the glaciers showed a flaky distribution for mapping. Visible comparison was suggested that
the extraction method used in this study had likely a strong ability to obtain the maximum length of
glaciers (Fig. 7a) and that its sensitivity to topography was lower than that of Machguth and Huss
(2014) (Fig. 7b). Both sets of glacier length extraction schemes were valid and there were large
differences only in a few glaciers or in certain types of glaciers, such as slope glaciers and ice caps.

Note that the comparative analysis results of the two lengths were relative, random samples were
limited, and it was difficult to accurately reflect the quality of the dataset in this study. Owing to
these limitations, the quality of the data must be determined again by secondary evaluation before
applying to specific regions. Additionally, the automatic extraction algorithm in this study is more
suitable for application to single-outlet glaciers, particularly valley glaciers; it is not suitable for ice
caps, flat-top glaciers, and tidal glaciers that are widely distributed in the Antarctic, sub-Antarctic,
northern Canadian Arctic, and other areas. Even if our algorithm can produce promising results,
accuracy remains a concern.

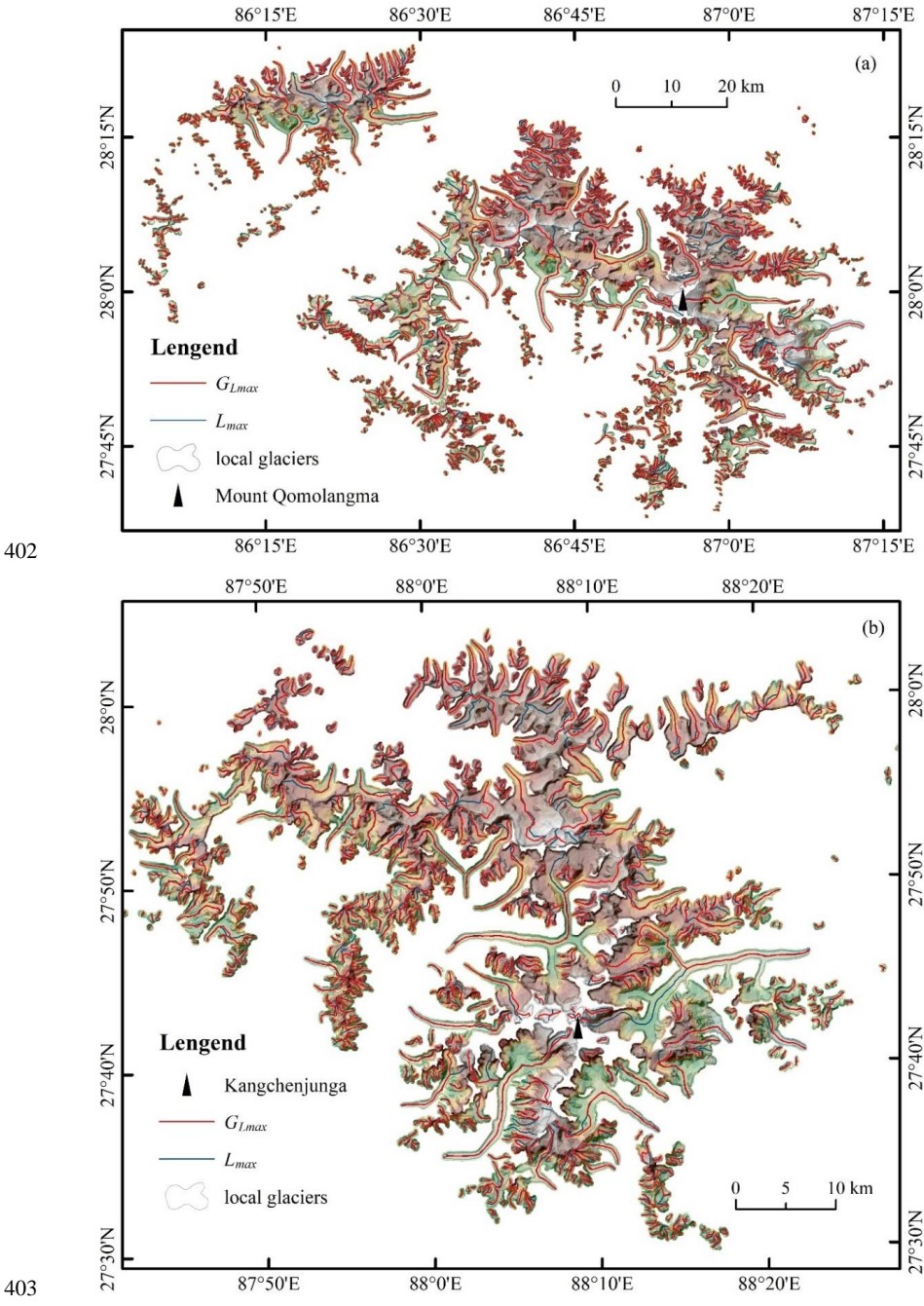



**Figure 7.** Visible comparison of the longest center lines calculated in this study and by Machguth
and Huss (2014). The figure shows two glacier-covered regions in the Himalayas, covering Mount
Qomolangma (**panel a**) and Kangchenjunga (**panel b**, the world's third highest mountain) and their
surrounding areas. The background is the DEM used for the calculation.



**5 Data availability**

Global glacier centerline dataset (*GGCLDS*), global glacier maximum length dataset (*GGMLDS*), and other relevant datasets are available at https://doi.org/10.11922/sciencedb.01643 (or https://www.scidb.cn/en/s/BRzaUf). All 14 sub-datasets of this dataset are listed in Table 5.

**Table 5.** Description of the members contained in this dataset.

| Acronym | Data format | Data volume | Description |
|---|---|---|---|
| *IGODS* | *.shp | 316 MB | Input glacier outline dataset |
| *GGEDS* | *.tif | 3.70 GB | Global glacier elevation dataset |
| *GGCLDS* | | 838 MB | Global glacier centerline dataset |
| *GGMLDS* | | 616 MB | Global glacier maximum length dataset |
| *GACLDS* | | 302 MB | Global glacier accumulation region length dataset |
| *GABLDS* | | 358 MB | Global glacier ablation region length dataset |
| *GOSRDS* | | 1.16 GB | Global glacier outline segmentation result dataset |
| *GLHPDS* | *.shp | 11 MB | Global glacial local highest point dataset |
| *GLPDS* | | 6.25 MB | Global glacial lowest point dataset |
| *GUGODS* | | 3.95 MB | Unsuccessful global glacier outline dataset |
| *FGODS* | | 119 MB | Global flawed glacier outline dataset |
| *GGECDS* | | 334 MB | Global glacier external contour dataset |
| *GGBMDS* | | 374 MB | Global glacier buffer mask dataset |
| *MHMLDS* | | 8.32 MB | The maximum length of Machguth and Huss in High Asia |

**6 Conclusions**

In this study, a new dataset on the centerline of global glaciers was constructed and the maximum length was calculated based on the global glacier inventory (RGI v6.0) and global glacier region DEM (*GGEDS*, composed of NASADEM, ASTER GDEM v3, and Copernicus DEM). In total, 198,137 glacier centerlines were generated, accounting for 99.74% of the total number of imported glaciers (*IGODS*) and 91.52% of the total number of the global glacier inventory. The comprehensive extraction accuracy of these glacier centerlines (*GGCLDS*) used in random self-assessment was 89.68%. The glacier length ($G_{Lmax}$) obtained in this study was generally approximately 10% longer than that of $L_{max}$ on average. Nevertheless, our method showed a stronger ability to obtain the maximum length, and we believe that the resulting errors were controllable. Furthermore, the preprocessing algorithm we designed accurately identified 10,764 erroneous glacier polygons from RGI v6.0, which formed the defective glacier dataset (*FGODS*).

A dataset containing 14 sub-datasets was generated through the above work, including two basic input datasets (*IGODS* and *GGEDS*), two key result datasets (*GGCLDS* and *GGMLDS*), four process datasets and six derived result datasets. Ice caps, nominal glaciers, and erroneous glacier polygons were eliminated from most sub-datasets in this study, accounting for approximately 8.25% of the total RGI v6.0. The poor status of these glacier polygons was not sufficient to support the automatic extraction of glacier centerlines, which needs to be improved in future work. Inevitably, there were some defects in the algorithm or datasets that need to be considered in future research. For instance, the glacial regions (R19 and R03) with the worst results were added to the dataset to prioritize data coverage integrity. It is worth noting that the global glacier DEM dataset (*GGEDS*), global glacier external outline dataset (*GGECDS*), and global glacier buffer mask datasets (*GGBMDS*) cover all glaciers in RGI v6.0. Accordingly, they will help researchers design more efficient automated extraction algorithms to produce datasets containing all types of glacier centerlines and lengths worldwide, which is our next goal.






**Appendix A:** Model parameters resulting from the Central Asia Glacier and extended to worldwide
calculations are listed in Table A1.
**Table A1.** Parameterization scheme for extracting global glacier centerlines.

| Par. | Description | Value (Levels 1-10) | Unit |
|------|-------------|---------------------|------|
| $P_1$ | Maximum distance between adjacent vertexes | 10 | m |
| $P_2$ | Buffer distance outside the glacier outline | 30 | m |
| $P_3$ | Threshold of accumulative flow | 5 - 8, 10, 20, 30, 50, 100, 200 | int × $10^3$ |
| $P_4$ | Length of the shortest auxiliary line | 10 - 19 | int |
| $P_5$ | Length of the longest auxiliary line | 2 - 11 | int |
| $P_6$ | Interval for searching the local highest points | 50, 60, 70, 80, 90,100, 200, 300, 400, 500 | count |
| $P_7$ | Matching tolerance of the vertexes of polyline | 0.2 ,0.2 ,0.5 ,0.5 ,1 ($L_5 - L_{10}$) | m |
| $P_8$ | Size of grid cell in Euclidean allocation | 1, 5, 15, 15, 30 ($L_5 - L_{10}$) | m |
| $P_9$ | Minimum distance between the adjacent $P_{max}$ | 10, 15, 30, 60, 120, 150, 200, 300, 400, 500 | count |
| $P_{10}$ | Smoothing tolerance of polylines | 5, 10, 15, 20, 30 ($L_5 - L_{10}$) | m |
| $P_{11}$ | Length threshold of the longest auxiliary line | 10190 | km² |

**Notes:** The calculation method for each parameter is detailed in Zhang et al. (2021). $P_{max}$ and $L$ refer to the local
highest points and grades of the glacier, respectively.

**Appendix B:** Comparison of longest centerlines calculated in this study and by Machguth and Huss
(2014) for all samples and the different first-order glacier regions of RGI v6.0. Linear regression of
the two lengths, histogram of length ratio ($R_r$), and box plots of $R_r$ for glaciers of different grades in
these regions were in Figure B1, B2, and B3, respectively.














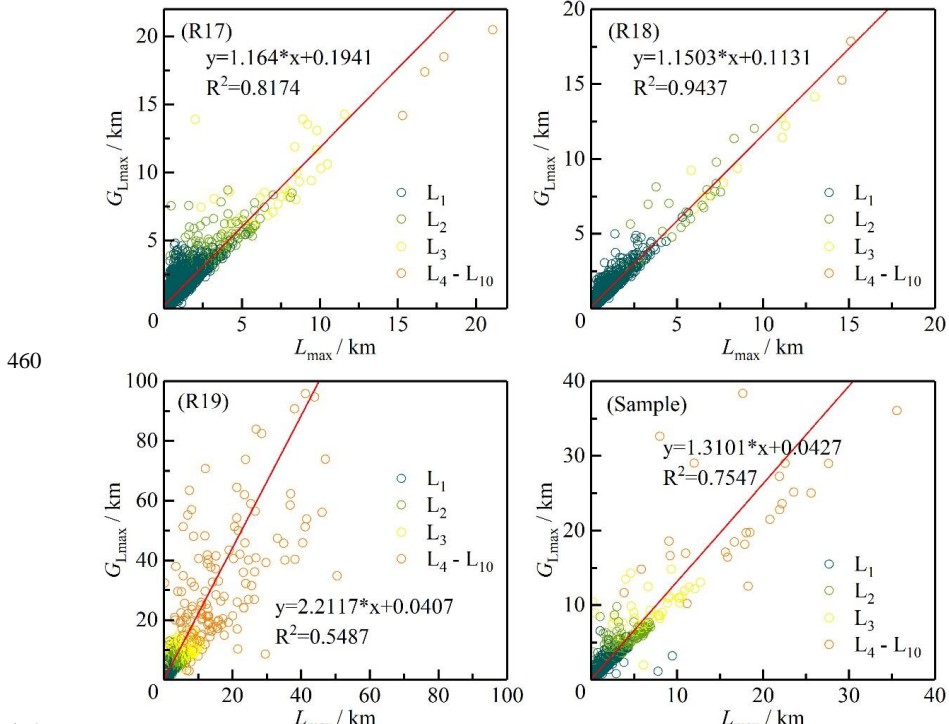

**Figure B1.** Linear regression in different glacier regions between glacier length ($G_{Lmax}$) calculated
in this study and glacier length ($L_{max}$) calculated by Machguth and Huss (2014).












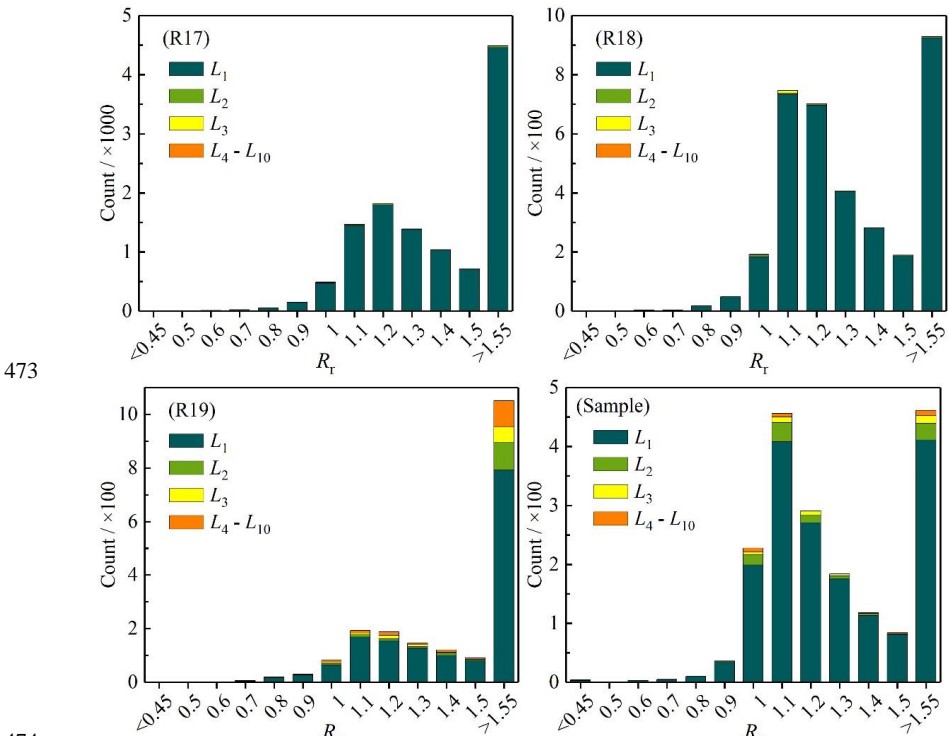


**Figure B2.** Histograms of the length ratio ($R_r$, $G_{Lmax}/L_{max}$) of distinct glacier grades in glacier-
covered regions and all samples.



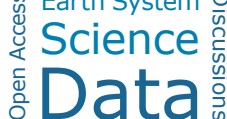









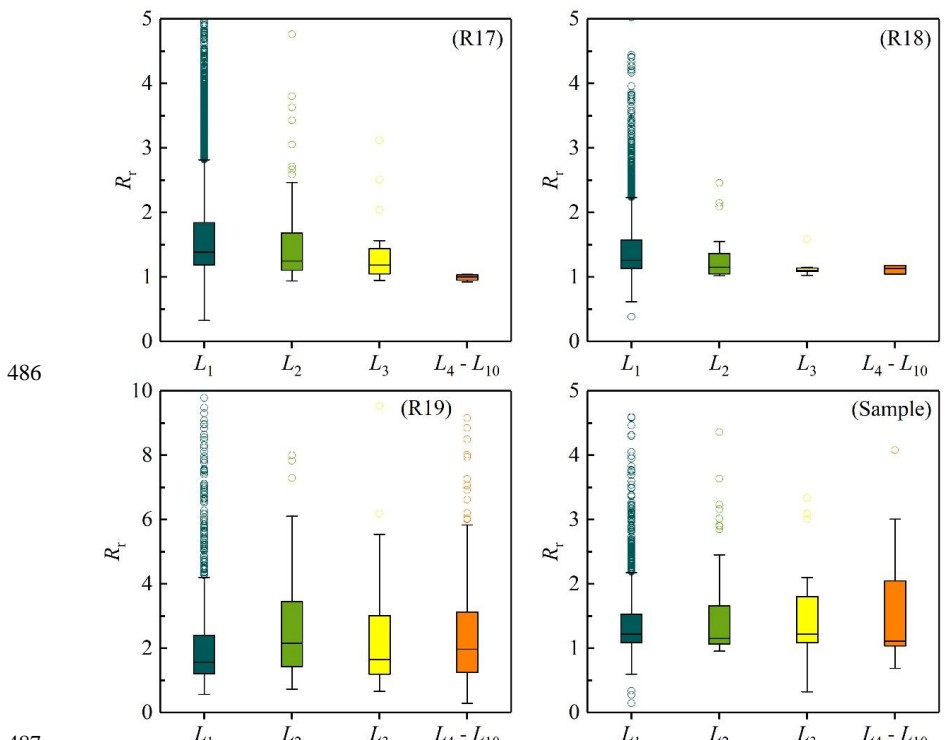

**Figure B3.** Box plots of length ratio ($R_r$, $G_{Lmax}/L_{max}$) of glaciers of distinct grades in every glacier-covered region and whole sample.



**Supplement.**

The Supplement consists of two parts: (1) 'GlacierCenterlines_Py27' (version 5.2.1), the updated automatic extraction tool of glacier centerlines in this study, which fixed some defects compared with version 5.2.0 (https://doi.org/10.5194/tc-151955-2021-supplement). (2) 'Other_parameters_T1.txt' is the parameter file for extracting the global glacier centerlines.

**Author contributions.**

All authors contributed to writing and editing the manuscript. DZ processed the data, performed all calculations, created all figures, and wrote most of the manuscript. SZ contributed significantly to the development of the analyses, figures, and writing. XY contributed to the development of the data production strategy and writing. GZ and WL contributed to the initial data production. SW participated in writing Chapter 4.

**Competing interests.**

The authors declare that they have no conflict of interest.

**Acknowledgments.**

The authors would especially like to thank GLIMS for releasing the RGI v6.0 (http://www.glims.org/RGI/randolph.html, last accessed: November 15, 2021), LP DAAC for releasing the NASADEM (https://lpdaac.usgs.gov/news/release-nasadem-data-products/, last accessed: November 17, 2021), METI and NASA for jointly releasing the ASTER GDEM v3 (https://lpdaac.usgs.gov/news/nasa-and-meti-release-aster-global-dem-version-3/, last accessed: November 17, 2021), and the European Space Agency (ESA) for providing the Copernicus DEM (https://spacedata.copernicus.eu/web/cscda/cop-dem-faq, last accessed: November 17, 2021). This work is not possible without the support of open-access data.

**Financial support.**

This research was funded by the Second Tibetan Plateau Scientific Expedition and Research Program (STEP) (grant number: 2019QZKK020109) and China National Natural Science Foundation (grant numbers: 41730751, 42171124).

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
