# Peer review of "A new global dataset of mountain glacier centerline and length"

_Earth System Science Data, 2022_

## Referee Comment (RC1)

**General Comments:**

Based on the glacier axis concept, Zhang et al. produced the global mountain glacier centerlines using the latest global glacier inventory and the digital elevation model data of corresponding glaciers. This research is challenging and heavy workload. The authors used the automatic checking algorithm to identify 10,764 glaciers with flawed outlines and mark the location of the defects. The centerline and related data of 198,137 worldwide mountain glaciers were automatically obtained by the compiled extraction tool, which is very important parameters for glacier research. The published datasets include not only the result data such as glacier centerline and maximum length, but also the key data such as DEM of glacier-covered and its buffer region and the glaciers of flawed outlines. The dataset has high quality, and the manuscript is generally well organized and written. The manuscript can be accepted after addressing my following comments.

**Specific comments:**

1. The manuscript mentioned that the automatic extraction tool does not support ice caps, nominal glaciers and some glaciers of flawed outlines, which accounts for 8.48% of the total number of worldwide mountain glaciers. I think it is necessary to add more details to the manuscript, including providing data users with possible approaches calculated the centerlines of these glaciers.

2. L243-L263: 100 random results for accurate evaluation in each region. Did you decide it yourself or refer to others? The number of glaciers in each region is different. Can a certain proportion be used to select centerlines, and the assessment results are possibly more convincing?

3. It is suggested to move the notes in Figure 1 after the caption of Figure 1.

4. L267-L271: Overall success rate or average success rate? How is it calculated?

5. L305-L315: Is it necessary to list a table to better understand?

**Technical corrections:**

L37 Add a space after 'changes'.

L77 '; the' -> ', their'.

L124 'However' -> 'Nevertheless'.

L195 'was' -> 'are'.

L225 'a glacier' -> 'glaciers'

---

## Author Response (AR1)

**Here are our responses to the two reviewers and to the community comment.**

**Please Notes:** Text in BLACK is the reviewer' comments and our responses are in BLUE. In addition, the notation used to locate the changes first defines the page number, then the line number(s). For example, **P4L15** means that the described modification to the manuscript can be found on the 15th line of the 4th page in the track-changes file.

**Response to Editor (Polina Shvedko)**:

Thanks for your helpful comments to improve this manuscript.

For now, we will proceed with your manuscript as submitted. However, please adjust your manuscript files before your next file upload (next round of revision or after acceptance) considering the following requirements:

1. Some supplement`s images are screenshots. If the screenshots were made by you, then no citation is required. If the screenshots were taken by others, then you must specify the source in the figure captions and the list of references.

Thanks for your reminder. The screenshots in the 'essd-2022-141-supplement' were all made by authors, and they are no conflict of interest.

2. Each DOI, no matter where, must be accompanied by a citation (e.g., Zhang and Zhang, 2022). However, the section "Data availability" in *.pdf manuscript file does not contain this information. Please add it for the next revision.

Thanks for your reminder. It has been added. **(P17L445)**

3. Please ensure that the colour schemes used in your maps and charts allow readers with colour vision deficiencies to correctly interpret your findings. Please check your figures using the Coblis – Color Blindness Simulator (https://www.color-blindness.com/coblis-color-blindness-simulator/) and revise the colour schemes accordingly.

Thanks for your reminder. All maps and charts in the new manuscript have passed the checks of the Coblis (Color Blindness Simulator). **(P3L97, P12L366, P18L483)**

**Response to reviewer 1**

Thanks for your helpful comments to improve this manuscript.

**General Comments:**

Based on the glacier axis concept, Zhang et al. produced the global mountain glacier centerlines using the latest global glacier inventory and the digital elevation model data of corresponding glaciers. This research is challenging and heavy workload. The authors used the automatic checking algorithm to identify 10,764 glaciers with flawed outlines and mark the location of the defects. The centerline and related data of 198,137 worldwide mountain glaciers were automatically obtained by the compiled extraction tool, which is very important parameters for glacier research. The published datasets include not only the result data such as glacier centerline and maximum length, but also the key data such as DEM of glacier-covered and its buffer region and the glaciers of flawed outlines. The dataset has high quality, and the manuscript is generally well organized and written. The manuscript can be accepted after addressing my following comments.

Thank you.

**Specific comments:**

- The manuscript mentioned that the automatic extraction tool does not support ice caps, nominal glaciers and some glaciers of flawed outlines, which accounts for 8.48% of the total number of worldwide mountain glaciers. I think it is necessary to add more details to the manuscript, including providing data users with possible approaches calculated the centerlines of these glaciers.

Thanks for your insights. We have added the new section 4.2.3 'Uncertainties and possibilities for improvement' in the manuscript, and the part focusing on explaining these problems is as follows:

For some glaciers that are not provided centerlines in this dataset, data users need to update the corresponding glacier outlines and could use the automatic extraction tool provided in this study to generate their centerlines, which involves the defective glacier outlines (*FGODS*), nominal glaciers and ice caps of the RGI v6.0. Specifically, the centerlines of the *FGODS* rely on the glacier outlines that meet the requirements of this study. These glacier outlines include glacier inventory data from other sources, or the *FGODS* that are repaired by some algorithms or manual process. Nominal glaciers are similar to *FGODS*, and also require users to obtain corresponding glacier outlines. Automatic approaches dividing ice caps from glacial complexes into individual glaciers are currently limited, and data users can only use their own criterion to divide ice caps and then use our tool to generate centerlines. **(P14L410)**

- L243-L263: 100 random results for accurate evaluation in each region. Did you decide it yourself or refer to others? The number of glaciers in each region is different. Can a certain

proportion be used to select centerlines, and the assessment results are possibly more convincing?

Thanks for your insights and suggestions. The number of input glaciers from different glacier regions in this study varies greatly: Iceland (R06) with 435 glaciers is the least and Central Asia (R13) with 52,858 glaciers is the most. Randomly selecting a certain proportion of centerlines in different regions for visual verification is a good approach, but not applicable to this study because the resulting gap of the sample size is probably orders of magnitude. Therefore, we decided to randomly select an equal number of centerlines from different glacier regions as the samples for visual verification. **(P8L246)**

- It is suggested to move the notes in Figure 1 after the caption of Figure 1.

    Thanks for your suggestion. It has been moved from the Figure 1 to the caption of the Figure 1. **(P3L97)**

[Figure]

**Figure 1.** Distribution of global glaciers, first-order glacier regions, and DEMs. The background is the global DEM grid (1°×1°) covered by NASADEM and GDEM. GDEM and COP DEM represent the ASTER GDEM v3 and the Copernicus DEM, respectively. **Notes:** R03: Arctic Canada, North; R05: Greenland Periphery; R06: Iceland; R07: Svalbard and Jan Mayen; R09: Russian Arctic; R12: Caucasus and Middle East; R13: Asia, Central; R14: Asia, South West.

- L267-L271: Overall success rate or average success rate? How is it calculated?

    Thanks for your insights. 99.74% is the overall success rate, which was calculated by the quantity ratio of the generated centerlines and all input glaciers. It has been modified to 'overall success rate'. **(P9L273)**

- L305-L315: Is it necessary to list a table to better understand?

    Thanks for your suggestion. This part describes the distribution of the flawed glacier outlines (*FGODS*) and ice caps in RGI v6.0, as shown in Table 2. We have added a reference to Table 2 in the section. **(P10L311**)

**Table 2.** Preprocessing results of different glacier regions and information of input datasets.

| Region | Region Name | Total | Ice Cap | Nominal glacier | Flawed glacier | Glacier input | DEM input |
|--------|-------------|-------|---------|-----------------|----------------|---------------|-----------|
| R01 | Alaska | 27108 | 0 | 0 | 704 | 26404 | NASADEM, GDEM |
| R02 | Western Canada and USA | 18855 | 0 | 0 | 1564 | 17291 | NASADEM, GDEM |
| R03 | Arctic Canada, North | 4556 | 650 | 0 | 47 | 3869 | COP DEM |
| R04 | Arctic Canada, South | 7415 | 953 | 0 | 63 | 6409 | NASADEM, GDEM |
| R05 | Greenland Periphery | 20261 | 1658 | 0 | 1547 | 17247 | COP DEM |
| R06 | Iceland | 568 | 133 | 0 | 1 | 435 | GDEM |
| R07 | Svalbard | 1615 | 144 | 0 | 12 | 1460 | GDEM |
| R08 | Scandinavia | 3417 | 0 | 4 | 75 | 3338 | NASADEM, GDEM |
| R09 | Russian Arctic | 1069 | 460 | 0 | 0 | 609 | GDEM |
| R10 | North Asia | 5151 | 5 | 116 | 136 | 4899 | COP DEM |
| R11 | Central Europe | 3927 | 0 | 2 | 76 | 3849 | NASADEM |
| R12 | Caucasus Middle East | 1888 | 0 | 339 | 2 | 1547 | NASADEM |
| R13 | Central Asia | 54429 | 1545 | 0 | 28 | 52858 | NASADEM |
| R14 | South Asia West | 27988 | 295 | 0 | 1946 | 25792 | NASADEM |
| R15 | South Asia East | 13119 | 289 | 0 | 4 | 12826 | NASADEM |
| R16 | Low Latitudes | 2939 | 0 | 0 | 724 | 2215 | NASADEM |
| R17 | Southern Andes | 15908 | 623 | 0 | 3828 | 11734 | NASADEM |
| R18 | New Zealand | 3537 | 0 | 0 | 0 | 3537 | NASADEM |
| R19 | Antarctic Subantarctic | 2752 | 419 | 0 | 7 | 2327 | COP DEM |
| -- | -- | 216502 | 7174 | 461 | 10764 | 198646 | -- |

**Note:** GDEM and COP DEM represent ASTER GDEM v3 and Copernicus DEM, respectively.

**Technical corrections:**

- L37 Add a space after 'changes'.

    Thanks for reminder. It has been modified. **(P1L37**)

- L77 '; the' -> ', their'.

    Thanks for reminder. It has been modified. **(P2L78**)

- L124 'However' -> 'Nevertheless'.

    Thanks for reminder. It has been modified. **(P4L128**)

- L195 'was' -> 'are'.

    Thanks for reminder. It has been modified. **(P7L201**)

- L225 'a glacier' -> 'glaciers'.

    Thanks for reminder. It has been modified. **(P8L229**)

**Response to reviewer 2 (Xin Wang)**

Thanks for your helpful comments to improve this manuscript.

**General Comments:**

I carefully reviewed the manuscript of "A new global dataset of mountain glacier centerline and length" submitted by Zhang et al. In this paper, the European allocation is applied to the automatic extraction of global mountain glacier centerline, which proved to be a feasible and reasonable approach. The manuscript includes a detailed description of data production, processing method and accuracy evaluation. The dataset is publicly available and its overall quality is good, which includes 14 sub-datasets including all input, process and result data. Besides of the *GGCLDS* and *GGMLDS*, I think the shared DEM (*GGEDS*), which was mosaicked with each glacier regions as units, is also a reasonable choice for relevant researchers to study. Overall, the manuscript is well-written with clearly structure. I think this manuscript can be considered for publication after some minor correction and technical comments have been addressed.

Thank you.

**Specific comments:**

- According to the automatic checking algorithm for the global glacier outlines in this study, my understanding is that the glacier polygons with defects only on the $P_{gec}$ are a high proportion in the *FGODS*, and they are probably to be supported by the automatic extraction tool. I suggest designing algorithms for this part of the *FGODS* to identify and repair them. The repaired glacier outlines are slightly distinguished from the RGI v6.0, so my suggestion is that their centerlines should be published as a supplementary dataset to increase the global coverage of this dataset.

Thanks for your insights and suggestions. Inspired of your comments, we designed a geometry-based algorithm to repair *FGODS* and provided data users with their centerlines in the form of a supplementary dataset, and corresponding codes and results are in sub-datasets *CODES* and *SUP_220707*. Generally, the glacier outlines with large coverage included in *FGODS* are mostly generated by automated extraction algorithm rather than manual vectorization, which are always jagged and have geometric flaws. The repair algorithm we designed is divided into five steps: (1) Searching the external contour of a glacier ($P_{gec}$), (2) identifying the closed polylines that exist the common vertices with the $P_{gec}$ and then deleting these closed polylines (if any), (3) iteratively searching the groups of closed polylines with common vertices within the glacier polygon, (4) traversing each group to delete the polylines except the longest closed polyline among them, and (5) merging remaining closed polylines and converting to a new glacier polygon. The comparison of three typical polygons of *FGODS* before and after processing by our repair algorithm are shown in Figure R1.

Note that the repairment of *FGODS* needs to consider two conditions of polygon geometry and glacier cover. The latter is very difficult and impossible to complete in current status, so the repair algorithm we designed only considers the former to prioritize the coverage of data users. The repaired glacier polygons are different from the RGI v6.0, such as slightly larger areas, local areas that may not match the actual glacier cover, etc. Therefore, we believe that the centerlines of these glaciers are not suitable for adding directly to the original dataset, nor for participating in the statistical analysis of the manuscript, and only provide data users in the form of supplemental datasets. **(P14L419)**

[Figure]

**Figure R1.** The schematic of the geometry-based algorithm to repair *FGODS*. Panels ($a_1$, $b_1$, and $c_1$) demonstrate the glacier polygons before repair, and panels ($a_2$, $b_2$, and $c_2$) are after repair.

- In general, the accuracy of 89.68% is acceptable for the results of fully automatic algorithm, but I am more concerned about the precautions for future readers to adopt the current dataset, the limitations of the dataset, and the possibility for improvement in the future. It is suggested to add a new chapter 4.2.3, focusing on the above problems.

    Thanks for your insights and suggestions. The new chapter 4.2.3 'Uncertainties and possibilities for improvement' has been added as follows:

**4.2.3 Uncertainties and possibilities for improvement **(P14L408)**

Although we compared the two current global length datasets, it is still difficult to accurately reflect the quality of the dataset in this study. For some glaciers that are not provided centerlines in this dataset, data users need to update the corresponding glacier outlines and could use the automatic extraction tool provided in this study to generate their centerlines,

which involves the defective glacier outlines (*FGODS*), nominal glaciers and ice caps of the RGI v6.0. Specifically, the centerlines of the *FGODS* rely on the glacier outlines that meet the requirements of this study. These glacier outlines include glacier inventory data from other sources, or the *FGODS* that are repaired by some algorithms or manual process. Nominal glaciers are similar to *FGODS*, and also require users to obtain corresponding glacier outlines. Automatic approaches dividing ice caps from glacial complexes into individual glaciers are currently limited, and data users can only use their own criterion to divide ice caps and then use our tool to generate centerlines. In addition, prioritizing the coverage of this dataset, we designed a geometry-based algorithm to repair FGODS and provided data users with their centerlines in the form of supplementary dataset, and corresponding codes and results can be seen in sub-datasets *CODES* and *SUP_220707*.

The automatic extraction algorithm in this study is more suitable for application to single-outlet glaciers, particularly valley glaciers; it is not suitable for ice caps, flat-top glaciers, and tidal glaciers that are widely distributed in the Antarctic, sub-Antarctic, northern Canadian Arctic, and other areas. In short, the uncertainties in this dataset come probably from the centerlines of some slope glaciers and the ice caps that are not identified in RGI v6.0, or a few centerlines with unpredictable quality due to the input data such as the incorrect glacier polygons, erroneous DEMs. In future work, better glacier inventory and more accurate DEM are useful for the improvement of centerline quality. On the other hand, optimizing the automatic glacier segmentation approach, DEM-based extraction algorithm of glacier feature lines and centerline trade-off algorithm are also probable ways to further improve the accuracy of glacier centerlines. In addition, it is probably beneficial to further clarify the type of each glacier in the glacier inventory for the estimates of centerline accuracy.

- If there are the qualified glacier outlines corresponding to the glaciers in the FGODS in the future, I hope to supplement their centerlines to this dataset in time.

  Thanks for your good suggestion. Since our dataset is in an open storage database, releasing of updated datasets are allowed at any time. We will update their centerlines to this dataset in time, if the qualified glacier outlines corresponding to these glaciers that are not provided centerlines are released in the future. **(P18L475)**

**Technical corrections:**
- L74 Delete 'of'.

  Thanks for reminder. It has been modified. **(P2L75)**

- L108 'better' -> 'smaller'.

  Thanks for reminder. It has been modified. **(P4L112)**

- L120 'ASTERGDEM'-> 'ASTER GDEM'

    Thanks for reminder. It has been modified. **(P4L124)**

- L208 total global mountain glaciers or total glaciers?

    Thanks for reminder. It has been modified to 'global mountain glaciers.' **(P7L212)**

- L362ã€• L488 Missing the name of horizontal axis.

    Thanks for reminder. The names of horizontal axes are all the 'Glacier level', and it has been added to the 21 corresponding figures. **(P12L366, P31L534)**

**Response to community comment 1(Wenfeng Chen)**

Thanks for your helpful comments to improve this manuscript.

Just a quick look. I downloaded the shared data and checked how it performed on the Tibetan Plateau indeed, and it was good overall. But there are many large glaciers with missing centerline data, such as the Karakorum region, is there any way to compensate for this?

Thanks for your insights. Mountain glaciers on the Tibetan Plateau involve three glacier regions: Central Asia (R13), South Asia West (R14), and South Asia East (R15). The dataset has 95.73% (91455/95536) coverage in High Asia (R13, R14 and R15). The glaciers that are not provide centerlines are mainly 1545 ice caps in R13 and 1946 defect glaciers in R14. Almost all glaciers in the Karakorum region are in R14, but the glacier outlines of R14 in the RGI v6.0 rely on automated extraction algorithm and they are generally jagged and have geometric flaws.

In the recent version of this dataset (https://doi.org/10.11922/sciencedb.01643), prioritizing the coverage of this dataset, we designed a geometry-based algorithm to repair *FGODS* and provided data users with 10676 centerlines of these glaciers in the form of supplementary dataset. Corresponding codes and results see sub-datasets *CODES* and *SUP_220707*. After the update, the coverage of this dataset in the mountain glaciers of High Asia has increased from 95.73% to 97.78%, and coverage in the R14 has increased from 92.12% to 99.01% (27711/27988). Unfortunately, nominal glaciers and ice caps still lacks the qualified sources of glacier outlines to calculate their centerlines.

For some glaciers that are not provided centerlines in this dataset, data users need to update the corresponding glacier outlines and use the automatic extraction tool in this study to generate their centerlines, which involves the defective glacier outlines (*FGODS*), nominal glaciers and ice caps of RGI v6.0. The *FGODS* and nominal glaciers are easy to generate centerlines as long as there are correct glacier outlines. However, automatic approaches dividing ice caps from glacial complexes into individual glaciers are currently limited, and data users only can use own criterion to divide ice caps and then use our tool to generate centerlines.

---

## Author Response (AR2)

**Response to Topical Editor (Min Feng):**

Dear Topical Editor,

Thank you very much for your valuable comments to improve this manuscript. We responded point by point to each comment as listed below, along with a clear indication of the location of the revision.

If you have any queries, please don't hesitate to contact us at the address below. Looking forward to hearing from you.

Thank you and best regards. Sincerely,

Dahong Zhang, Gang Zhou, Wen Li, Shiqiang Zhang \*, Xiaojun Yao, Shimei Wei Email: Shiqiang Zhang (zhangsq@lzb.ac.cn) and Dahong Zhang (zhangdh\_yx@163.com)

**Please Notes:** Text in BLACK is the comments and our responses are in BLUE. In addition, the notation used to locate the changes first defines the page number, then the line number(s). For example, **P4L15** means that the described modification to the manuscript can be found on the 15th line of the 4th page in the track-changes file.

**Comments to the author:**

I would like to thank the authors for submitting the revision, which I believe has addressed the previous review comments. However, there are still many ill-expressions and confusing sentences in the manuscript. I think that the authors need to carefully check and improve the writing of the manuscript before it is accepted for publication. I would also encourage the authors to ask for help from native English speakers or language professionals.

Thanks for your decision and good suggestions. We try our best to improve the manuscript followed by your suggestions. The whole text has been polished by native English-speaker of Insiderofscience Company (Fig. R1). We believe that all possible questions are clarified and it is useful and friendly to future readers.